# The Critical Factors Affecting the Implementation of Corporate Governance in Indonesia: A Structural Equation Modeling Analysis

## Suldja Hartono *, Mochammad Al Musadieq, Kusdi Rahardjo and Tri Wulida Afrianty

Faculty of Administrative Science, Brawijaya University, Malang 65145, Indonesia
* Correspondence: suldjahartono@gmail.com

**Abstract:** The concept of corporate governance has become a topic of global discussion since The New York Stock Exchange crashed on 19 October 1987, when many multinational companies listed on the New York Stock Exchange experienced large financial losses. CG was a preventive measure and increased investor confidence in the company. CG implementation is influenced by isomorphisms, such as organizational structure and the external environment in the form of regulation, competition, and culture. In Indonesia, the quality of CG implementation still contributes to the country's economic growth. Weak CG implementation is due to the adoption of the western system immediately. It arises due to a high ownership structure, government intervention, underdeveloped capital markets, and weak law enforcement. This study aims to examine and analyze more deeply about factors that cause CG not to be inadequate to develop properly in Indonesia. Private and state-owned companies in the East Kalimantan Industrial Estate are the research samples for the SOR modelling exploration method. The SOR (stimulus-organism-response) model is a novelty in identifying CG implementation. Identification of the model to obtain a structural model is carried out by using PLS-SEM (partial least square structural equation modelling) through an institutional approach. The results found that the organizational structure and national cultural environment strongly influence CG implementation through the mediation of organizational structure. The contribution to understanding the national cultural environment in CG implementation efforts will be driven by organizational structure. Comprehensively, this study describes other factors such as organizational culture, environment competition, and the environment mediated by organizational structure. The national cultural environment mediated by organizational culture did not significantly affect CG.

**Keywords:** CG implementation; SOR model; organizational structure

## 1. Introduction

Corporate governance has become a topic of global discussion since The New York Stock Exchange Crash on 19 October 1987, when many multinational companies listed on the New York Stock Exchange experienced huge financial losses. Implementing Corporate Governance (CG) is considered an anticipation for company executives to hide losses with financial engineering aimed at revamping performance and financial reports (Pramono 2011). One of these preventive measures is contained in the CG guidelines, which require an independent commissioner to form an audit committee tasked with overseeing financial management and observing the reporting process to encourage reliable financial reports so that it can increase investor trust.

CG issues in the first generation were dominated by conflicts of interest between principals and agents because of the separation between ownership and control of a modern company. Conflicts of interest between principals and agents arise when a company grows larger so that the company's management that was originally held by the owner (owner-manager) must be handed over to professionals. In this case, the issue that is considered dominant is the need for a mechanism to ensure that the management (agent), who is

a worker in a company that owns capital (principal), will manage the company in the interests of the owner (Berle and Means 1932).

In the second generation, CG issues lead to conflicts of interest between strong majority and weak minority shareholders (La Porta et al. 1998). According to La Porta et al. (1998), the application of CG in a country is influenced by the condition of legal instruments in protecting the interests of various parties related to the company, especially minority owners. A legal system that is not conducive and is not yet in favor of the public interest can result in a strong conflict between strong majority shareholders and weak minority shareholders, so it has the potential to damage the country's economic system as a whole.

In Indonesia, CG has begun to be widely applied to companies since the 1997 monetary crisis. The application of CG has become a vital need for state-owned companies (BUMN) and private companies (individuals) as a fundamental value for increasing the welfare and sustainability of companies (Dwiridotjahjono 2009). However, CG principles still needed to be fully implemented during the early 21st decade, and CG has yet to show significant progress in Indonesia. It is illustrated by the annual Credit Lyonnais Securities Asia (CLSA) (2018) survey regarding the evaluation of CG implementation in Asia Pacific countries. Indonesia still ranks 12th, which only fulfils 34% of the 100 assessment indicators. Fundamental problems causing hampered implementation of CG include, among others (Pramono 2011), the practice of companies financed by banks belonging to business groups, as well as short-term loans from abroad, shareholder domination, the ineffective performance of regulators and financial institutions, and weak protection of creditors and investors.

One of the efforts to improve CG implementation in Indonesia is appreciating BUMN and BUMS companies. Nevertheless, these efforts still have not increased the professionalism of the company. Winner companies are still involved in disciplinary cases, such as PT PLN (BUMN) in the Riau-1 PLTU bribery case and PT Garuda Indonesia in the aircraft procurement bribery case. It has become a concern of researchers to identify why the implementation of CG in Indonesia has not been successful even though the regulations, instruments, and tools are good. Indreswari (2006) and Allen and Gale (2000) stated that the low performance and efficiency of BUMN occurred due to the monopoly of certain sectors, which encouraged abuse of authority. Morgan (2006) mentioned that competition could replace external CG management to improve the company's quality so that the organization will be closely related to the environment. As a stimulus factor, the environment has elements that will affect the organization (Lubis and Huseini 1987).

Isukul and Chizea (2015), in research conducted in Nigeria, South Africa, and Egypt, stated that a good and conducive environment would support the development of CG in each country, and regulations have a major impact on CG (La Porta et al. 1998). Stimulus in regulation will demand compliance, while stimulus in competition focuses on increasing organizational value on marketability and improving the quality of human resources (Udayasankar et al. 2008). CG implementation as a response is expected to increase the professionalism of managers. Sharma and Joseph (2003) explained that the implementation of CG will only be successful with the support of professionalism from parties related to the company. The value of professionalism has several attributes, namely self-confidence, service, confidentiality, competence, contract, community, caring, and commitment (Spitzech and Hansen 2010). Hall (1968) found that professionalism was not correlated with authority and the existence of regulations and procedural specifications. However, professionalism is positively correlated with a division of labor, impersonality, and compensation. Dawuda (2010) also explains that CG can be used as an antidote to corruption and a proven control in several countries to ensure good governance is running. Dawuda's research found that CG can form competent and professional personnel if the governance structure has been properly implemented and implemented.

Furthermore, Wahyudin and Solikhah (2017) found that understanding the importance of good corporate governance (GCG) has already grown in Indonesian businesses. The listed businesses that took part in CGPI Awards from 2008 to 2012 always see an

improvement in both number and quality. Accounting-based financial performance, such as return on assets, return on equity, and earnings per share, are impacted by the CG rating of Indonesian go-public businesses. The Indonesian stock market does not immediately respond to CG implementation rating, so it has yet to be able to accelerate the company's growth soon. In this study, to find out how the environment in the form of regulation, competition, and societal culture influences organizations in implementing CG, the stimulus-organism-response (SOR) model is used Mehrabian and Russell (1974). The SOR paradigm is applied to three levels of interrelated variables. This model implies that human behavior can be better understood as an interactive process in which environmental events (stimuli) are perceived and processed cognitively by individuals in organizations that lead to a type of behavior (response). D. J. Campbell (1985) confirms that the SOR model can be used to explain individual behavior towards their work in organizations. Figure 1 gives a framework chart for using the SOR model (Mehrabian and Russell 1974) in this research.

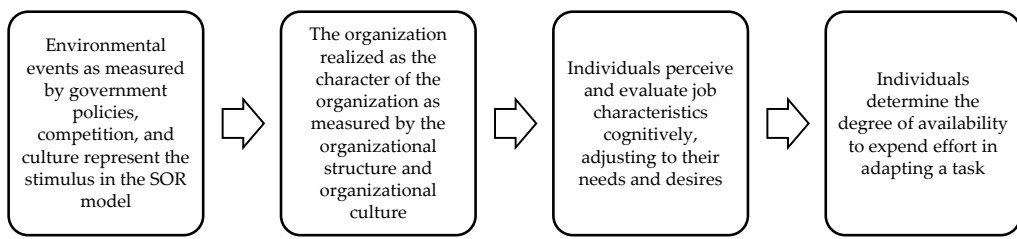

**Figure 1.** SOR Model Thinking Framework in Research.

The SOR model framework is a novelty in this study. The environment is a variable that influences the organization in realizing good CG implementation. Environmental influence analysis uses appropriate institutional theory in explaining, understanding, and measuring organizational behavior. DiMaggio and Powell (1983) argued that organizations will operate in similar environments and adopt the same structure and culture. In previous research, Udayasankar et al. (2008), Su (2006), and Gallego-Alvarez and Pucheta-Martinez (2019). The institutional theory used to analyze environmental and organizational influences is not applied to the SOR model framework, which is considered to be used to measure behavior in this study as the perception of managers in implementing CG. Udayasankar et al. (2008) measured regulatory environment variables and competition in the organization. Su (2006) measured culture, and Gallego-Alvarez and Pucheta-Martinez (2019) measured regulatory (coercive), competitive (mimetic), and cultural (normative) environments based on institutional theory to measure their effect on firms in disseminating information.

The implementation of CG in Indonesia has yet to increase the professionalism of managers, even though there are sufficient institutional regulatory instruments from the government. Using the theory of individual behavior, the various problems of abuse, and problems that exist, it can be shown that the perception of CG has not been able to form an attitude that builds professional behavior. The SOR behavior change model and institutional theory are expected to explain the influence of regulation, competition, and culture on organizational culture and structure, which will affect the implementation of CG. This combination of different models and theories will capture the interactive influence of the regulatory environment, competition, and culture on organizational culture and structure so that CG implementation can shape manager professionalism.

This research was conducted in the Kaltim Industrial Estate (KIE) area considering the material's mastery level and the relatively controlled environment. The KIE area also has companies whose shareholders are controlled by the government (BUMN) and the private sector, both individuals and foreign investment (PMA), so this research can examine more deeply about the research subject. The motivation to conduct this research is to answer why CG has not developed properly in Indonesia (ACGA 2014; Indreswari 2006). This study

compares the influence of employees' perceptions of state-owned and private companies on implementing CG quantitatively using an institutional isomorphism approach. The research conducted expected to provide an empirical contribution to the application of the stimulus-organism-response model in obtaining a solution to the problem of why the application of CG in companies still cannot increase professionalism; provide an empirical contribution in testing that CG will develop and be closely related to the environment in which the company is located; provide an empirical contribution in testing that government with its regulations, industry with competition, and culture with corporate ethics and norms are three environmental dimensions that greatly influence the implementation of CG; and provide an empirical contribution to the implementation of CG principles as a measurement indicator of CG implementation. This research also provides empirical contributions in the application of institutional theory models because institutional theory can be considered capable of explaining, understanding, and measuring organizational behavior, and contributing empirically to testing the theory used to explain the process of adaptation in institutional practice within organizations called an isomorphism. This research enhances the SOR model framework as a novelty and implements it on companies whose shareholders are controlled by the government.

The research consists of introduction of the research, theoretical concepts, research hypothesis, the methodology used for this research, the measurement of variables and the results of the research, the discussion of the results, and concludes by providing research limitations, managerial implications, and discussion material for further research.

## 2. Theoretical Concepts and Hypothesis Development

Since it became a global discussion in 1987, corporate governance has been used to anticipate company executives so as not to hide the company's losses. The Asian crisis is suspected because of weak CG implementation (Johnson et al. 2000). Research organized by ADB (2000) stated that the weakness of CG in Asia arose due to a high level of ownership structure, government intervention, underdeveloped capital markets, and weak law enforcement. Furthermore, research on CG has been carried out by Paniagua et al. (2021), who examines how CG and ownership structures relate to the financial performance of firms. Several factors, including organizational culture, organizational structure, regulatory environment, competitive environment, and national cultural environment, influence the implementation of CG in a company.

Organizational culture is defined as organizational practices that express the values of that organization. Deal and Kennedy (1982) and Peters and Waterman (1982) are some of the works that popularized organizational culture. The popularity of the organizational culture literature during the 1980s appears to have responded to the decline in corporate performance in the United States vis-a-vis firms in Japan. Academics seek to explain this decline by relying on culture as the main factor. In another study (Hofstede 2001), the focus on culture shifts to the power one has over CG where culture becomes a strong explanatory variable.

Contrary to previous arguments regarding a direct relationship between organizational culture and performance, this study's conceptual framework hypothesizes that organizational culture is not directly related to performance. However, the relationship occurs through internal CG. Semenov (2000) argues that " . . . simple models linking culture to performance no longer match the knowledge that academics have developed regarding culture's role in organizational analysis. There is a need for a better understanding of the relationship between culture and impact on organizations".

**Hypothesis 1 (H1).** *Organizational culture will have a significant effect on CG implementation.*

Semenov (2000) compared the CG systems of industrialized western countries and argued that cultural scores explained differences in CG in 17 Western countries better than any other economic variable listed in the literature. Lin (1976) supports this argument and

demonstrates an intercultural theory of CG systems based on the dimensions of cultural values that link shareholder structure and self-dealing arrangements, insider trading, and disclosure. Since organizational culture studies, internal CG are rare, this study seeks to fill a gap in the literature. This argument is similar to Schwartz and Davis (1981), who stated that 'company culture has a major impact on a company's ability to realize goals and plans . . . '.

Organizational structure describes how work is divided, grouped, and formally coordinated. In the context of CG, Blau and Schoenher (1971) defines CG in its broadest sense as "the totality of the legal, cultural, and institutional arrangements that determine what public companies can do, who controls, how those controls are implemented, and how the risks and rewards of activities are allocated. In contrast, Shleifer and Vishny (1997) give a narrow definition of CG, who state that CG deals with 'how financial suppliers guarantee themselves a return on their investment". Similarly, the Cadbury Committee of Financial Aspects of Corporate Governance defines CG as 'the system that directs and controls companies' (Cadbury 1992). Nguyen (2022) found the fact that, in corporate governance at banks, there is a difference between bank stability and the effectiveness of audit committees that depend heavily on the soundness of each bank and the institutional quality of each country. On the other hand, Dang and Nguyen (2021) found that internal corporate governance is significantly related to future stock risk. These different definitions reflect CG's perspectives and areas that must be addressed. This broad definition captures not only the function of a corporate governance structure or organ, but also its external environment, which consists of social influences, government regulations that regulate companies, and labor and capital markets.

**Hypothesis 2 (H2).** *The organizational structure will affect the implementation of CG as a control mechanism.*

Meanwhile, the narrow definition only places CG as a company's business affairs, including the company's internal structure and management processes. The organization as a structure will greatly influence how CG is implemented. According to Walsh et al. (2011), organizational structure will influence internal and external mechanisms and governance to ensure that CG will be implemented.

**Hypothesis 3a (H3a).** *The regulatory environment is expected to significantly affect the implementation of CG by mediating organizational culture.*

Regulation is the management of decisions that are made very complex following a set of rules made by the government and were in force at that time. To their needs, regulations are made according to the context. The regulatory environment requires the compliance of the various parties involved to behave by the established rules of the game so that organizational goals can be achieved effectively. In terms of CG in Indonesia, the government stipulates regulations that must be complied with by companies in Indonesia (Morgan 2006). Gallego-Alvarez and Pucheta-Martinez (2019) states that the regulatory environment will face formal and coercive pressures to comply with social standards within organizations. This coercive power is closely related to regulatory agencies that have the power to sanction companies (e.g., legal mechanisms). For J. L. Campbell (2006), Coercive pressure is closely related to the main regulatory instruments that can sanction companies, such as legal and enforcement mechanisms. Larrinaga (2007) views this type of coercive isomorphism as involving regulations that encourage the disclosure of ecological information, guarantee mandatory compliance, or threaten future regulation. Coercive pressure is usually associated with governments and regulatory agencies. This pressure is closely related to the main regulatory instruments that can sanction companies, such as legal and enforcement mechanisms. J. L. Campbell (2006) supports that companies will behave more responsibly by conducting their activities in an institutional environment with greater coercive pressure and where the legal system is oriented towards protecting stakeholder interests.

**Hypothesis 3b (H3b).** *The regulatory environment is suspected to influence the implementation of CG by mediating the organizational structure.*

The institutional theoretical notion is that the institutional environment can greatly influence the development of formal and informal structures in an organization, often greater than market forces and pressures (Lounsbury 2005). The institutional theory addresses elements of social structure in a deeper and more resilient way: there is a need to consider the processes by which structures, including norms, rules, routines, and schemes, become institutionalized as authoritative parameters or guides for social behavior (Scott 2004). Following the philosophy and logic of this theory, it can be argued that one of the main influences responsible for effective CG compliance within a country is the existence of institutions that can compel organizations to adopt and implement transparent and fair CG practices (Judge et al. 2008). Greenwood et al. (2008) argues that coercive isomorphism occurs because organizations tend to be motivated to avoid sanctions. In research, La Porta et al. (2002) found that best practices in CG can only thrive in the presence of a good legal and regulatory framework. For a CG framework to be effective, legal entities and regulators must be sound so that investors can rely on them when they enter into contractual agreements.

Intense product market competition forces management to improve financial performance and make the best decisions for the future since failure to do so is likely to result in bankruptcy and job loss. Well-managed companies will take over the market from poorly managed companies. The competition will help bring out the best performance from the management team and discipline management. In Allen and Gale's (2000) model, competition is a substitute for external CG mechanisms, particularly the market for firm control.

**Hypothesis 4a (H4a).** *The competitive environment will affect the implementation of CG by mediating organizational culture.*

The competitive environment is considered to influence the discipline of organizations by eliminating inefficient organizations (Udayasankar et al. 2008). Holmstrom (1982) considers that the competitive environment makes monitoring more efficient on the culture of corruption by managers. Udayasankar et al. (2008) and Gallego-Alvarez and Pucheta-Martinez (2019) classify the competitive environment as an isomorphism through a mimetic process. Scott (2001) identifies from mimetic processes due to cognitive institutional influences. He argues that the mimetic institutional perspective through resource dependence as one of the reasons that can explain the effect of competition must be mediated by a productive and useful organizational culture to be able to make CG implementation effective.

**Hypothesis 4b (H4b).** *The competitive environment will influence the implementation of CG by mediating the organizational structure.*

Udayasankar et al. (2008) explained that the competitive environment has a negative effect on CG if the organization has a complex structure. This argument is based on the competitive advantage that arises from CG, which acts as a driving force for organizations to improve CG implementation. However, Udayasankar et al. (2008) demonstrated that, as perceptions of the competitive environment increase with high organizational structure complexity, it weakens CG rather than enhances it. Hatch and Cunliffe (2013) emphasizes that the type of structure is the most important thing in the organization. This opinion is based on the organizational structure that will encourage or restrain an innovation from being implemented. The organizational structure is a boundary that opens up various possibilities. These constraints create the possibility of choice and action. Without any restrictions, possible action will not exist (D. J. Campbell 1985).

**Hypothesis 5a (H5a).** *The national cultural environment will influence the implementation of CG by mediating organizational culture.*

Wibowo (2008) emphasizes that organizational culture is a resource that produces competitive cultural advantage. A company that ultimately allows companies to achieve better results. Hitt et al. (2001) conducted studies that study the relationship between organizational culture and CG, which illustrates that the explanatory power of organizational culture becomes very important when the national cultural environment also supports organizational culture as an invisible resource in generating competitive advantage. The Indonesian context is no exception because organizational culture is a long-lasting resource and provides better company performance. It is because the cultural environment will shape organizational culture, which is valuable, rare, inimitable, and irreplaceable. The unique nature of the cultural environment that shapes organizational culture will differ greatly from country to country (Barney 1986). Related to the importance of organizational culture to CG, culture needs to be studied thoroughly to reveal its role in CG (Schein 1992). From a theoretical point of view, CG is thought to help prevent scandals, fraud, and other potential problems that can damage a company. A company with a good CG image will enhance the company's reputation. Semenov (2000) states that organizational culture significantly impacts a company's ability to realize goals and plans.

**Hypothesis 5b (H5b).** *The national cultural environment will influence the implementation of CG by mediating the organizational structure.*

Khan and Law (2018) states that the cultural environment is composed of values and beliefs and is the programming of the collective mind. The cultural environment system is a set of values, attitudes, and ways structurally and historically developed and shared. The cultural environment will directly or indirectly affect the organization in terms of organizational design, work design, and organizational rewards. In terms of how good CG implementation emphasizes the existence of structural variables where the cultural environment mechanism is translated into a structure within the organization. Feng (2017) states that the complexity, formalization, and centralization of decisions will greatly affect the implementation of CG in a company. According to him, studying the organizational structure is a way to focus on maximizing CG contribution. Gallego-Alvarez and Pucheta-Martinez (2019) state that the cultural environment that influences the organization is the basis of normative isomorphism, where the organizational structure will adapt to the norms, values, and orders that distinguish one society from others throughout the world. Thus, the cultural environment guides the organization in shaping its structure.

## 3. Methodology

This research was conducted with a quantitative approach using a questionnaire or survey method, which consists of an explanatory survey with a correlational design and a descriptive survey. The research population is state-owned and private companies in the Bontang Industrial Estate (KIE) Kaltim Industrial Area in Lok Tuan, North Bontang. The BUMN cluster has a population of 289, while the private cluster has a population of 189, bringing the total population in this study to 407 people. A sample survey was conducted from a population of 407 people to test the instruments to be used. Facts were found in the field that line 3 managers have very low awareness. They do not even know about CG, which is the subject of research. With these considerations in mind, line 3 managers who are operators and technical employees were removed from the list of populations to be targeted.

The target population in this study was 199 respondents, with a total of 144 SOE respondents and a total of 55 private respondents. The sampling technique in this research used purposive sampling by considering the size and representativeness of the population.

The sample limit measurement used the Slovin formula ([Bordens and Abbott 2011](#)), as in Equation (1).

$$n = \frac{N}{1 + Ne^2}$$

(1)

where *n* is the sample size, *N* is the population size, and *e* is the error rate.

The proportion of the BUMN sample is 72.3% of the total minimum sample using the Slovin formula, while private companies are 29.3%. Thus, the minimum sample of BUMN is 95 respondents and the private sector is 39 respondents. Sources of research data come from primary data and secondary data. Primary data were obtained by: (1) a questionnaire survey; (2) an interview survey; and (3) non-reactive methods and available statistical data. Collecting primary data using an instrument in the form of a questionnaire consisting of closed questions using intervals and open questions is used to obtain a more comprehensive picture. Open questions use a ratio scale to be coded and analyzed using statistical tools. In contrast, secondary data were obtained by policy documents, statistical documents, or monographs and reporting documents issued by state-owned and private companies. The policy documents used in this study are (1) Law Number 40 of 2007 concerning Limited Liability Companies; (2) BUMN Law Number 19 of 2003; (3) Regulation of the Financial Services Authority Number 21/POJK.04/2015 concerning the Implementation of Public Company Governance Guidelines; and (4) SOE Minister Regulation Number Per 01/MBU/2011 concerning the Implementation of Good Corporate Governance (GCG).

In this research, there are three variable concepts, including the Regulatory Environment, the impact on professionalism will be used the stimulus-organism-response (SOR) model of [Mehrabian and Russell](#) ([1974](#)). Regulatory Environment (X1), Competition Environment (X2), Cultural Environment (X3), Organizational Culture (M1), Organizational Structure (M2), and Implementation of Corporate Governance (Y1). Scaling indicators—variable response indicators using interval scales with scores 1–7, which means that the value one will be worse and the value seven will be better for assessing the variables' attributes ([Nachmias and Nachmias 1987](#)). Data will be said as good and quality if using quality measurement instruments. A quality instrument is an instrument that has a reliability of a measure and validity or validity of a measure. The variable reliability test method used the Pearson Product Moment Correlation Reliability method and Cronbach's Alpha. Pearson Product Moment Correlation, to measure the strength of the relationship between variable X and variable Y, and be used to determine the validity of an instrument for several interval data. Validity test of the measure to find out how well the indicators represent the variables following the operational definition of the variable: the better the suitability, and the higher the validity of the measurement ([Newman et al. 2013](#)). Validity test of the criteria level carried out in research by testing and calculating the Pearson Correlation coefficient between each indicator with a total score of all indicators.

The research data analysis method using descriptive data analysis and partial least squares analysis (PLS-SEM) makes it possible to simultaneously test the relationship between multiple exogenous and endogenous variables to explore and predict the relationship between latent variables because the theory is undeveloped or weak. Partial Least Square (PLS) is a multivariate statistical analysis that can estimate/test the research model simultaneously both the relationship between variables or between variables and their measurement items with the aim of predictive studies [Hair et al.](#) ([2006](#)).

Structural analysis in PLS-SEM in this study can be explained in Figure [2](#). Latent variables are represented in a circle, while the latent variable forming indicators are represented with long ovals. Arrows represent the relationship between latent variables and latent variables with indicators. In PLS-SEM, the relationship is always shown as a one-way arrow. The stages of analysis in PLS-SEM are explained by [Sudarmanto](#) ([2005](#)). In summary, the PLS-SEM evaluation of the reflective latent variable measurement model and the structural equation model can be seen in Table [1](#).

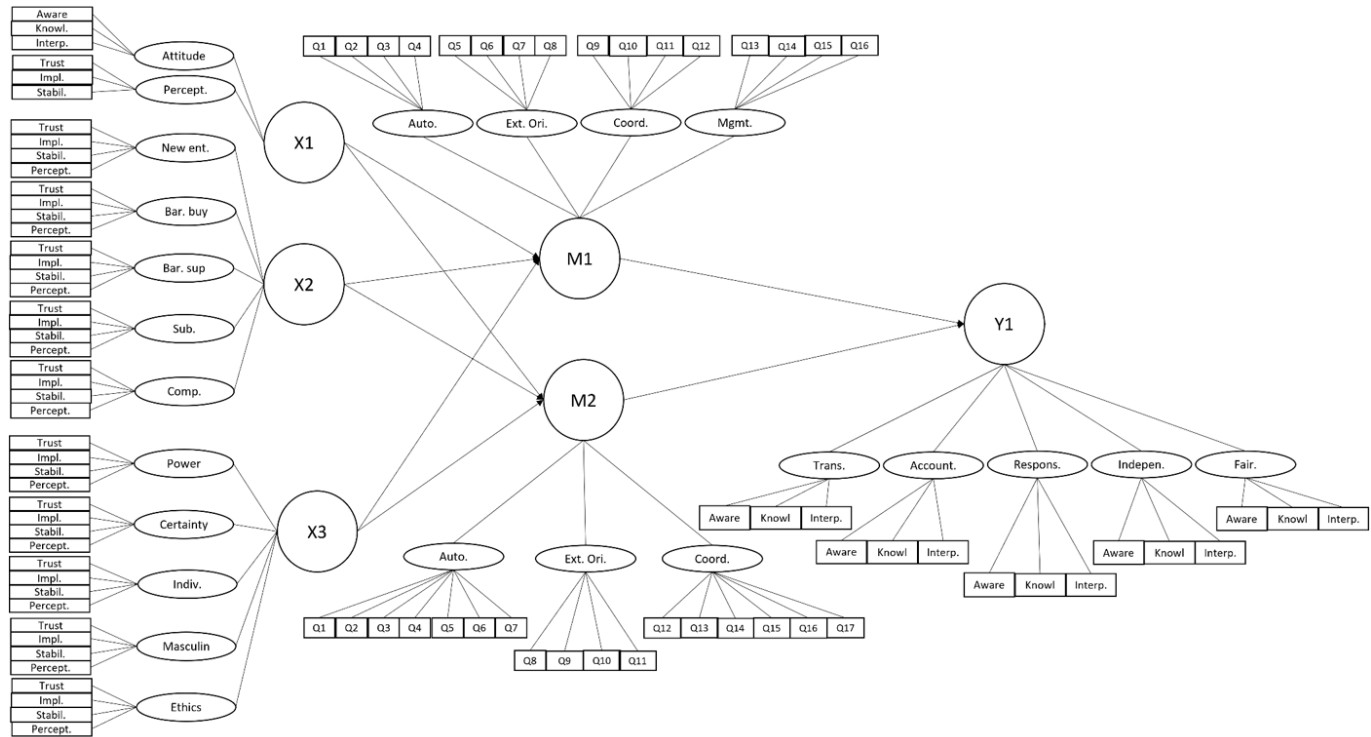

**Figure 2.** Research Structure Equation Model.

**Table 1.** Evaluation of the results of the PLS-SEM model.

| Evaluation | Indicators | Appropriateness |
|---|---|---|
| Outer Model | 1. Indicator reliability 2. Discriminant validity 3. Internal consistency 4. Convergent validity | 1. Outer loading must be ≥0.7 for the theory test and 0.5–0.7 for exploratory research 2. Cross-loading the indicator variable to the latent variable must be greater in value than the other latent variables. Fornell-Larcker, each latent variable must be greater than the correlation between latent variables. 3. Composite reliability ≥ 0.7 for the theory test and ≥0.6 for exploratory research. Cronbach's alpha ≥ 0.7 for the theory test and ≥0.6 for exploratory research 4. Average Variance Extraction (AVE) must be greater than 0.5 |
| Inner Model | 1. The coefficient of determination ($R^2$) 2. The significance and magnitude of the structural model coefficients | 1. In general $R^2$ value ≥ 0.75 is good 2. Significant |

## 4. Results

Analysis of the influence between variables was carried out by analytical methods PLS-SEM (partial least square structural equation modelling) with the aim of predictive or exploratory studies through the development of structural models (Hair et al. 2006). The PLS model consists of measurement and structural models. The measurement model uses second-order factors with variables hierarchically measured by dimensions, and several measurement items further measure these dimensions. The estimation of second-order factors is then carried out using the repeated indicator approach (first-order factor stage), followed by the two-stage approach for evaluating the causality between variables and dimensions (second-order stage) after obtaining a valid and reliable model (Hair et al. 2006). This research focuses more on second-order analysis.

### 4.1. The Measurement Model at the Variable Level Evaluation

The measurement model was evaluated at the second-order factor level, which measures the quality of the measurement model of the relationship between the variables and their dimensions. The results of the measurement model at the variable level are presented in Table 2. The quality of the measurement model is seen from the Loading Factor (LF) $\geq$ 0.70, Composite Reliability (CR) $\geq$ 0.70, and Average Variance Extracted (AVE) $\geq$ 0.50, as well as an evaluation of discriminant validity, which is the Fornell-Lacker Criterion, which is the AVE root above the correlation between variables.

**Table 2.** Validity and Reliability of Research Variable Dimensions.

| Variable | Dimension | Loading Factor | Composite Reliability | Average Variance Extracted |
|---|---|---|---|---|
| Regulation (X1) | Attitude | 0.952 | 0.951 | 0.906 |
| | Perception | 0.952 | | |
| Competition (X2) | New arrival | 0.741 | 0.894 | 0.630 |
| | Bargaining buyer | 0.849 | | |
| | Bargaining supplier | 0.707 | | |
| | Substitution | 0.774 | | |
| | Competitor | 0.884 | | |
| National Culture (X3) | Equality | 0.560 | 0.861 | 0.558 |
| | Certainty | 0.838 | | |
| | Individualism | 0.744 | | |
| | Masculinity | 0.772 | | |
| | Ethics | 0.791 | | |
| Organizational Culture (M1) | Autonomy | 0.574 | 0.892 | 0.679 |
| | External Orientation | 0.892 | | |
| | Coordination | 0.901 | | |
| | Human Resources | 0.883 | | |
| Organizational Structure (M2) | Complexity | 0.803 | 0.877 | 0.704 |
| | Formalization | 0.873 | | |
| | Decentralized/Centralized | 0.841 | | |
| Corporate Governance (Y1) | Transparency | 0.729 | 0.912 | 0.675 |
| | Accountability | 0.883 | | |
| | Responsibility | 0.808 | | |
| | Independency | 0.796 | | |
| | Fairness | 0.883 | | |
| | Effectiveness | 0.968 | | |
| | Efficiency | 0.971 | | |
| | Responsibility | 0.966 | | |

The results show that the regulatory environment variable is measured by two dimensions, namely attitudes and perceptions where there is a very strong relationship between the two dimensions with an LF of 0.952 each. It can be caused by employees/managers having good attitudes and perceptions regarding PJOK, regulations, and laws. LF value greater than 0.7 indicates that the variable indicator has a high level of validity. The variable indicator must be eliminated or removed from the model if the value is smaller. The level of strength or truth is still weak (Ardiansah 2017). The competitive environment variable is measured by five accurate dimensions, where the most dominant dimensions reflecting the competitive environment were competitors (LF = 0.884) and bargaining power of buyers (LF = 0.849).

In contrast, bargaining power of supplier has LF = 0.707, which is good but still needs improvement. The national cultural environment is measured by five valid dimensions with LF, where the most dominant dimensions are certainty with LF = 0.838 and ethics with LF = 0.791. On the other hand, the equality dimension has the lowest LF (0.560), indicating that equality in a national culture still needs improvement.

Organizational culture variables are measured by four valid dimensions, where the very dominant dimensions are coordination (LF = 0.901) and external orientation dimensions (LF = 0.892). In contrast, the autonomy dimension still needs improvement with the lowest loading factor (LF = 0.574). Organizational structure dimensions are measured by three valid dimensions where, overall, there is a strong relationship between the dimensions of complexity, formalization, and decentralization/centralization in measuring organizational structure variables. However, the formalization dimension has the highest LF (0.873), indicating that the organizational structure's most important dimension is formalization. The CG variable has five valid dimensions, and the most important/dominant dimensions are accountability and fairness, with each LF value of 0.883. CG looks stronger in the dimensions of accountability and fairness.

This measurement model has CR values above 0.70 and AVE above 0.50 for each variable. It shows that the dimensions that measure the variables are reliable/reliable or consistent in measuring each variable (Mulyana et al. 2017). The content of dimensional variations in the research variables is more than 50%, indicating that the variables have good convergent validity. These results also indicate that structural testing of the influence between variables can be carried out with the support of a good measurement model.

Discriminant validity was carried out in the PLS analysis of this study to ensure that each dimension/item of focus measurement measures the variables it measures that are related or unrelated (Farrell and Rudd 2009). The method used in evaluating discriminant validity is the Fornell-Lacker criterion, namely, the root of the AVE variable is greater than the correlation between variables. The results of discriminant validity measurements are presented in Table 3. Based on the processing, it can be seen that all the roots of the AVE variable are higher than the correlation with other variables, which indicates that the evaluation of discriminant validity is fulfilled.

**Table 3.** Discriminant validity.

| Variable | National Culture | Organizational Culture | Corporate Governance | Competition | Regulation | Organizational Structure |
|---|---|---|---|---|---|---|
| National Culture | 0.747 | | | | | |
| Organizational Culture | 0.621 | 0.824 | | | | |
| Corporate Governance | 0.151 | 0.212 | 0.822 | | | |
| Competition | 0.421 | 0.360 | 0.214 | 0.794 | | |
| Regulation | 0.061 | 0.043 | 0.266 | 0.097 | 0.952 | |
| Organizational Structure | 0.536 | 0.539 | 0.325 | 0.338 | 0.124 | 0.839 |

Based on the stages of the model testing process, the VIF value of each variable being tested needs to be calculated to avoid multicollinearity, so that the estimated parameter values and standard errors are not biased. From the data processing presented in Table 4, the variables of regulation, competition, and national culture in influencing organizational culture and organizational structure show a VIF value of <5 or less than the tolerance limit, according to Hair et al. (2006). It can be concluded that there is no high multicollinearity among the variables of regulation, competition, and culture. Likewise, for organizational culture and organizational structure in influencing corporate governance, VIF value < 5.

**Table 4.** Multicollinearity testing.

| Variable | Organizational Culture | Organizational Structure | Corporate Governance |
|---|---|---|---|
| Regulation | 1.010 | 1.010 | |
| Competition | 1.223 | 1.223 | |
| National Culture | 1.216 | 1.216 | |
| Organizational Culture | | | 1.410 |
| Organizational Structure | | | 1.410 |

*4.2. Structural Model Evaluation*

The results of testing the model hypothesis as a whole based on each hypothesis statement are presented in Table 5. Hypothesis analysis used the parameter path coefficient value from −1 to 1. The hypothesis is accepted if the T-statistic is less than the *p*-value and the *p*-value < 0.05.

**Table 5.** Results of testing the structural model hypothesis.

| Hypothesis | Hypothesis Statement | Standard Path Coefficient | T-Statistic | *p* Value | Result |
|---|---|---|---|---|---|
| H1 | Organizational culture → Corporate Governance | 0.052 | 0.542 | 0.588 | Hypothesis Rejected |
| H2 | Organizational structure → Corporate Governance | 0.296 | 3,419 | 0.001 | Hypothesis Accepted |
| H3a | Regulatory environment → CG Implementation; Organizational Culture Mediation | 0 | 0.02 | 0.984 | Hypothesis Rejected |
| H3b | Regulatory environment → CG Implementation; Organizational Structure Mediation | 0.025 | 1.076 | 0.283 | Hypothesis Rejected |
| H4a | Competition Environment → CG Implementation; Organizational Culture Mediation | 0.006 | 0.407 | 0.684 | Hypothesis Rejected |
| H4b | Competition Environment → CG Implementation; Organizational Structure Mediation | 0.038 | 1.397 | 0.163 | Hypothesis Rejected |
| H5a | Cultural Environment → CG Implementation; Organizational Culture Mediation | 0.03 | 0.534 | 0.594 | Hypothesis Rejected |
| H5b | Cultural Environment → CG Implementation; Organizational Structure Mediation | 0.141 | 3.087 | 0.002 | Hypothesis Accepted |

The analysis results generally show that all hypotheses have a positive path coefficient direction with different significance for each variable. The hypothesis is not accepted, meaning that the relationship between variables is not significant, which indicates that, every time one variable changes, it does not significantly increase changes in other variables, namely organizational culture and CG implementation; regulatory environment and CG implementation through the mediation of organizational culture and organizational structure; environment competition and CG implementation through the mediation of organizational culture and organizational structure; as well as the cultural environment and CG implementation through the mediation of organizational culture. Conversely, there is a significant influence on the organizational structure variable on CG and the organizational environment and CG implementation mediated by organizational structure.

## 5. Discussion

The use of the term corporate governance (CG) has increased when factors involve sustainable investor confidence, shareholder activity, increased social responsibility, and sustainable organizational development. CG is a system of arrangements that directs and controls the company to increase value for all relevant stakeholders (Blau and Schoenher 1971). CG development is an indicator that cannot be separated from the level of investor trust. The increased intention on the influence factor of CG quality becomes important in economic development. In this study, the factors that have a significant or positive influence on the implementation of CG, namely the organizational structure and the national cultural

environment through the mediation of the organizational structure, while other variables do not have a significant effect even though they have a positive direction or in the other word, it has a very weak effect on the implementation of CG.

Organizational structure has a significant effect on CG implementation because each structure will direct the behavior of managers in implementing CG in their daily work. Cosset et al. (2016) mentioned that companies with good CG on average have better labor productivity and cost efficiency, and can make acquisitions that can increase company value, meaning that the organizational structure is good. According to Monks and Minnow (2004), CG is a structural mechanism intended to guarantee checks and balances that reflect the long-term sustainability of an organization. In addition, a significant influence on the implementation of CG was also identified in the national cultural and environmental factors through the mediation of the organizational structure, which is like the results of the study DiMaggio and Powell (1983), Scott (2001), and Hofstede (1991). In these studies, it can be interpreted that there is ruler control and scientific selection in the formation of the organizational structure in CG implementation. The dominant political power, or what Hofstede calls the elite (Hofstede 1991), is to apply the norms and standards of the national culture as a model of organizational structure and policies, which then apply years later without being questioned or forming a culture (Bebchuk and Roe 1999).

The success of national cultural variables in influencing CG implementation by mediating organizational structure provides strong support for the argument that isomorphism embedded in national culture will influence CG implementation strategies by establishing a strong organizational structure that aligns with company goals. Contribution to the understanding of the national cultural environment will be driven by the organizational structure, which is considered a residue of cultural norms in that country. This finding provides a theoretical implication that, in an institutional approach, companies try to seek legitimacy in society by conforming to societal norms and culture. Consistent with DiMaggio and Powell (1983), who state that organizational structural conformity is driven by institutional strength that is not related to efficiency in implementing it.

Furthermore, some factors have a weak influence on CG implementation, namely organizational culture, regulatory environment, regulatory environment mediated by organizational culture and organizational structure, competitive environment mediated by organizational culture and organizational structure, and national cultural environment mediated by organizational culture. The CG system initiating from the west will deal with organizational culture, a company resource that has formed the order and is a competitive advantage for companies, making CG a foreign system tasked with controlling and directing management (Hitt et al. 2001). In line with the findings above, Semenov (2000) compared the CG system in western countries, and it turns out that knowledge of these countries is still low. Insufficient knowledge of CG significantly impacts a company's ability to realize CG implementation (Schwartz and Davis 1981). With a lack of knowledge and understanding of CG in the work environment, the work culture in the company where the research is conducted separates the work culture that has been formed from the CG culture.

According to Tabalujan (2002), the regulatory environment in Indonesia requires a fundamental change to the legal culture so that people can become more law-abiding and principle-abiding. Such conditions are needed so that legal instruments and supporting institutions can function optimally following appointed objectives based on the legal culture and culture of the Indonesian people. Traditional cultural values are more dominant than legal-formal institutionalized legal rules (Lukviarman 2004). Johnson et al. (2000) state that the loss and non-functioning of organizational culture does not give life to the existing legal/regulatory system because culture refers to the attitudes, values, and opinions held by members of the organization regarding its implementation. The non-existence of an unsupportive organizational culture in companies makes CG implementation even more complicated. He believes that organizational culture is not a significant determinant of CG implementation in the companies studied. These results are inconsistent with the findings of previous studies such as Wilderom and Van den Berg (2005). However, a weak

relationship with the determination of organizational culture was recorded in research by Wibowo (2008). Arogyaswamy (1987) claims that organizational culture in a regulatory environment is not always crucial in determining CG implementation. The mediation of organizational structure in this variable is due to the regulatory environment in its legal products by Bebchuk and Roe (1999), considered to have the dominant power to regulate the structure of the company. This legal force is not always made by officials who side with the public and are not influenced by important groups, but it also has implications for other possible perspectives with the position of the two poles of the shareholding and stakeholder perspectives.

Economic changes influence the competitive environment in the context of company competitiveness, which is expected to lead to the implementation of CG with a management control system. However, Wibowo (2008) indicates that organizational culture is not a determinant of the significance of the company's internal CG. Specifically, Arogyaswamy (1987) claims that culture and competitive environment are not always crucial in determining the success of CG implementation in a company. The application of organizational structure mediation on competitive environment variables also does not significantly affect the implementation of CG. This result is contrary to previous studies such as Pfeffer and Leblebici (1973), Nickell (1996), and Porter (2008). However, the weak influence of the competitive environment on CG mediated by organizational structure was noted in the study by DiMaggio and Powell (1983). The emphasis is on research by Polder et al. (2009) regarding the importance of implementing the best CG in every company in a globally competitive environment as protection against potential risk threats. However, the behavior of CG implementation cannot be predicted even though a company restructuring has been carried out as a limitation in carrying out management. Meanwhile, implementing CG practices is still just to check the compliance box. DiMaggio and Powell (1983) stated that self-awareness and self-interest are very important in improving the development of organizational structures.

The national cultural environment and organizational culture do not strongly influence each other in implementing CG because they explicitly comply with Semenov's (2000) argument that national culture limits variance in organizational culture. However, Hatch also stated that culture also relies on differences besides relying on similarities. It means that not all values are accepted collectively but can be rejected collectively (Hatch and Cunliffe 2013). Specifically, Gerhard (2008) argues that organizational culture does not have to determine national culture in designing and executing management strategies and practices so that national culture acts as a strong boundary. The decision to be unique, as long as the risks and challenges are properly understood and considered, can often offer potential competitive advantages. Therefore, it should not reduce the space for freedom and differentiation. It is appropriate that Hatch and Cunliffe (2013) states that national culture may not be able to answer the challenges faced at any moment. It is the basis for identifying when national culture limits organizational culture and when it is possible to use it.

Different contexts in the form of the legal and regulatory environment, cultural environment, and business patterns (competition) that are predominantly adhered to in a country are the main factors that deserve consideration in identifying CG implementation systems and models. Thus, the effectiveness of governance tools does not depend on the number of existing regulations but depends heavily on the regulatory environment in the form of instruments and law enforcement in a country. It is what Tabalujan (2002) claimed allegedly caused the failure of CG implementation in companies that foreign technical assistance funds mostly assisted. In his research, it was explained that one of the reasons for the non-functioning of law in developing countries, especially in Indonesia, is due to the neglect of the cultural factors of the Indonesian people. The implication is that regulatory issues are not the only dominant factor influencing CG implementation. Other factors interact in Indonesia that influence CG implementation, such as environmental factors on the effectiveness of implementation and its supporting institutions (Lukviarman 2004).

From a formal legal standpoint, Tabalujan (2002) believes that Indonesia already has a fairly complete set of laws. What is needed is a fundamental change to the legal culture so that people can become more law-abiding and principle-abiding. Conditions like this are needed so that legal instruments and supporting institutions can function optimally so that they are following the stated goals. Thus, it can be said that traditional cultural values play a more dominant role than formally institutionalized regulations.

## 6. Conclusions

This research serves as a basis for identifying factors influencing this research. This research can summarize the modelling of CG implementation in Indonesia based on an institutional approach to three types of isomorphism, which emphasizes the institutionalization of CG implementation. This study confirms the cultural environment as a normative isomorphism from three perspectives of isomorphism in the institutional approach that influences CG implementation. The influencing normative isomorphism is based on national culture mediated by organizational structure. Ruler control and scientific selection occur in the formation of the organizational structure in the implementation of CG. The success of national cultural variables in influencing CG implementation by mediating organizational structure provides strong support for the argument that isomorphism embedded in national culture will influence CG implementation strategies by establishing a strong organizational structure that aligns with company goals. Contribution to understanding the national cultural environment in CG implementation efforts will be driven by the organizational structure, which is considered a residue of cultural norms in that country. The PLS-SEM analysis method with an institutional approach can describe measurement and structural factor models that influence CG on various variables. As a result, this study found that CG practices are strongly influenced by organizational structure and the national cultural environment mediated by organizational structure. In addition, it was also confirmed that sound CG practices that pay attention to the cultural aspects of certain countries would have an optimal impact.

There is no research without limitation. This research used quantitative methods with limited variables. Further research using qualitative methods is needed to deepen the research results regarding the implementation of CG. Future research is expected to be able to carry out limitations or restrictions on focus variables so that research results are sharper and more in-depth about CG in Indonesia, as well as being input into further analysis and application of regulations related to improving the quality of CG in Indonesia. This research can become the foundation for developing a research model with other variables to identify CG implementation.

**Author Contributions:** Conceptualization, S.H.; methodology, S.H.; software, S.H.; validation, S.H.; formal analysis, S.H.; investigation, S.H.; resources, S.H.; data curation, S.H.; writing—original draft preparation, S.H.; writing—review and editing, S.H., M.A.M., K.R. and T.W.A.; visualization, S.H.; supervision, M.A.M., K.R. and T.W.A.; project administration, S.H.; funding acquisition, S.H. All authors have read and agreed to the published version of the manuscript.

**Funding:** This research received no external funding.

**Informed Consent Statement:** Not applicable.

**Data Availability Statement:** Not applicable.

**Acknowledgments:** Thank you to the supervisors and all parties involved.

**Conflicts of Interest:** The authors declare no conflict of interest.

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
