# Peer review of "The Critical Factors Affecting the Implementation of Corporate Governance in Indonesia: A Structural Equation Modeling Analysis"

_economies, doi:10.3390/economies11050139_

Round 1
Reviewer 1 Report
Title: The Critical Factors Affecting the Implementation of Corporate Governance in Indonesia: A Structural Equation Modeling Analysis
After reviewing this article, I think it is potential for publication but the authors should revise as comments below:
- In the introduction, the authors should emphasize the motivation and the contribution of the research. Are there any differences from previous studies?
- At the end of the introduction, the authors should introduce the structure of the paper.
- In the literature review, the studies reviewed are very old, so the authors should review and update recent studies relating to corporate governance. I suggest the authors review and cite some recent studies such as Dang and Nguyen (2021); Nguyen (2022a); Paniagua et al. (2018)… (see reference).
- In the discussion, the authors should discuss the findings of the study in relation to the Indonesian context, and what support or support does this study provide to previous studies.
- In the conclusion, the authors need to present research limitations
- There are some typos and grammatical errors, you must check it again carefully.
References
Dang, V. C., & Nguyen, Q. K. (2021). Internal corporate governance and stock price crash risk: evidence from Vietnam. Journal of Sustainable Finance & Investment, 1-18. doi:10.1080/20430795.2021.2006128
Nguyen, Q. K. (2022a). Audit committee structure, institutional quality, and bank stability: evidence from ASEAN countries. Finance Research Letters, 46, 102369.
Paniagua, J., Rivelles, R., & Sapena, J. (2018). Corporate governance and financial performance: The role of ownership and board structure. Journal of Business Research, 89, 229-234.
Reviewer 2 Report
1. Originality and significance of contribution - contribution to existing knowledge;
The paper is dealing with a promising research idea (the critical factors affecting the implementation of corporate governance in Indonesia).
The originality and significance of contribution is given by the investigation of the selected variables (regulation, competition, national culture, organizational culture and organizational structure) towards corporate governance.
The authors used an already existing methodology /model (PLS-SEM), which was adapted to the pursued objectives.
2. Appropriateness of Abstract as a description of the paper;
The abstract shows, concisely, the objectives and results of the paper.
3. Adequacy of the literature review;
The introduction manages to familiarize the reader with the context of the chosen research topic.
The introduction is accompanied by a presentation of the relevant and adequate literature of the chosen topic. The authors briefly present the most important works in the chosen field.
4. Organization and readability;
The paper is logically and specifically organized, following a typical research paper structure (introduction, theoretical concepts and hypothesis development, methodology and data, model results, discussion, conclusion, references).
There is a mistake in numbering the paper structure (the methodology and the results have the same number, 3).
The paper needs minor proofreading, grammar and spelling correction.
5. Soundness of methodology, analysis, and interpretation;
The research methodology is correctly chosen and adapted for the pursued objectives.
There is a variable (Y2), which appear in figure 2 and it is not explained in the paper.
The results are correctly interpreted and are relevant for the chosen research objectives.
However, there is an important concern that needs to be raised. The paper is written in a way that it looks like a well-performed methodological exercise, but the data description is missing. In order to verify the applying methodology and the results, the data verification is a critical phase. The data description (line 321-325) must be extended by using as many appendices needed. The primary data (questionnaire survey, interview survey, and non-reactive methods and available statistical data) and the secondary data (policy documents, statistical documents or monographs and reporting documents issued by state-owned and private companies) should be described in detail.
This important concern needs to be addressed by the authors.
6. Evidence supports conclusions.
The conclusions correctly reflect the approaches from the model results and discussion section.
Round 2
Reviewer 1 Report
this paper can be accepted
Author Response
Thank you for your feedback and appreciation about our paper.
Reviewer 2 Report
Soundness of methodology, analysis, and interpretation;
The authors partially succeeded to address our concerns. They described the secondary data (policy documents), but the description of the primary data (questionnaire survey, interview survey, and non-reactive methods and available statistical data) is insufficient. The structure of the questionnaire survey and the interview survey, together with the centralization of all the answers should be added in the appendices or should be sent to editors and reviewers separately from the paper.
This important concern needs to be addressed by the authors.
Round 3
Reviewer 2 Report
The authors succeeded to address our concerns.